# Molecular Testing as Triage in Cervical Cancer Screening: Economic Evaluation Using Headroom Analysis

**DOI:** 10.3390/cancers17040612

**Published:** 2025-02-11

**Authors:** Kelly M. Castañeda, Karin M. Vermeulen, Antoinette D. I. van Asselt, Ed Schuuring, G. Bea A. Wisman, Marcel J. W. Greuter, Geertruida H. de Bock

**Affiliations:** 1Department of Epidemiology, University Medical Center Groningen, University of Groningen, 9700 RB Groningen, The Netherlands; k.m.vermeulen@umcg.nl (K.M.V.); a.d.i.van.asselt@umcg.nl (A.D.I.v.A.); g.h.de.bock@umcg.nl (G.H.d.B.); 2Department of Health Sciences, University Medical Centre Groningen, University of Groningen, 9700 RB Groningen, The Netherlands; 3Department of Pathology & Medical Biology, University Medical Centre Groningen, University of Groningen, 9700 RB Groningen, The Netherlands; e.schuuring@umcg.nl; 4Department of Gynecologic Oncology, University Medical Centre Groningen, University of Groningen, 9700 RB Groningen, The Netherlands; g.b.a.wisman@umcg.nl; 5Department of Radiology, University Medical Center Groningen, University of Groningen, 9700 RB Groningen, The Netherlands; m.j.w.greuter@umcg.nl

**Keywords:** Cost analysis, Uterine cervical neoplasms, Human papillomavirus viruses, Molecular diagnostic

## Abstract

This study presents a novel perspective on estimating the maximum reimbursable price for molecular triage testing to achieve cost-effectiveness. It employs SiMCerC, a new micro-simulation Markov model designed for Dutch cervical cancer screening. It explores different sensitivity and specificity combinations, analyzing their effects on colposcopy referrals, screen-detected CIN2+, false positives, and overall costs. This research highlights the advantages of incorporating molecular testing for self-sampled hrHPV-positive women, leading to reduced follow-up visits and improved resource utilization in the screening program.

## 1. Introduction

Cervical cancer is the fourth most common female cancer worldwide, accounting for around 660,000 new cases and 350,000 deaths in 2022 [1]. Most of these are caused by the sexually transmitted infection high-risk human papillomavirus (hrHPV) [2]. Though such infections are usually transient and disappear without intervention, a small proportion will persist and may progress to cervical intraepithelial neoplasia grade 2 (CIN2) or 3 (CIN3) [3]. If left untreated, these lesions have the potential to progress to cervical cancer [3].

Effective hrHPV-based primary screening programs have been developed to replace the less effective cytology-based primary screening [4]. Compared with cytology, hrHPV tests have a higher sensitivity (90% vs. 73%) for detecting CIN2 or worse (CIN2+, including CIN2, CIN3, and cancer) [5]. Furthermore, hrHPV screening-based programs offer two additional benefits: first, the option of self-sampling, which might increase participation rates, and second, reducing screening intensity by extending intervals for hrHPV-negative women due to the high sensitivity of the test [5,6,7,8]. However, due to the low specificity of the hrHPV test, many transient hrHPV infections are detected, resulting in unnecessary colposcopy referrals, overtreatment, anxiety, and increased costs [7]. To address this, cytology has been used as a triage test in hrHPV-positive women, helping to discriminate those at high risk of having CIN2+ from those at low risk [7].

A limitation of using cytology as triage is that self-sampled material cannot be used for cytological examination, meaning that these women require an extra visit to the general practitioner (GP) [9,10]. In the Netherlands, the use of self-sampling devices has increased from 4.1% in 2017 to 44% in 2023 among all participants; however, around 10% to 20% of women who use self-sampling and test positive for hrHPV do not return for cytology [11,12]. Therefore, research into additional triage tests is ongoing [13], and although the results are inconsistent, molecular-based tests (e.g., genotyping and methylation) have shown promise for detecting CIN2+ among HPV-positive women, with sensitivities and specificities of around 70% and 80%, respectively [14,15,16]. These molecular tests can also be carried out on the same sample as the hrHPV primary screening test and can be collected by the GP/clinician or as self-collected samples [13]. Although an economic evaluation in the Netherlands found that the use of DNA methylation reduced the cost per screen compared with cytology in the short term [17], the maximum cost of molecular-based testing for cost-effectiveness in population-based cervical cancer screening, including long-term costs and outcomes, has not been extensively evaluated.

In this study, we assessed the potential commercial value of molecular triage testing, given its clinical performance, in an organized cervical cancer screening framework. To do so, we developed, validated, and applied a microsimulation Markov model for cervical cancer screening (the SiMCerC model) to compare screening with a molecular test and cytology as triage. This model focuses on the ability of triage testing to distinguish CIN2+ from CIN1 or less, assuming a histological progression from HPV to CIN to cancer.

## 2. Materials and Methods

### 2.1. Description of the Model and Input Variables

The description of the model is presented in detail in Appendix A. Briefly, SiMCerC is a microsimulation Markov model used for cervical cancer screening. It comprises two parts: one models the natural history of cervical cancer, while the other explores several screening scenarios. The model has six states (Appendix A), the duration of each cycle in the model is 1 year, and transition probabilities between states were obtained from the literature (Table 1, Appendix A) [3,18,19]. The model simulated a non-HPV-vaccinated cohort from birth to death, considered age-dependent overall probabilities of death, and allowed for different screening scenarios (see below). Outcomes of interest were the numbers of life-years (LYs), colposcopy referrals, screen-detected CIN2+, and women referred to colposcopy with a false-positive test.

### 2.2. Scenarios

#### 2.2.1. Base Screening Scenario

The framework of the Dutch cervical cancer screening program (2021), which relies on cytology for triage, was used to simulate the base screening scenario [29]. The program invites women aged 30, 35, 40, 50, and 60 years for primary hrHPV testing, but only invites those aged 45, 55, and 65 years if they are hrHPV-positive or did not participate at ages 40, 50, and 60 years, respectively. All input parameters for compliance with the cervical cancer screening program were based on national monitoring data in the Netherlands (Monitor 2019) (Table 2) [12].

Participation in primary screening was determined by age from among the participants in each screening round, estimating that 9% use self-sampling and the remainder attend primary screening by a GP [12]. If a primary test for hrHPV is positive, a cytology triage test is usually performed on the same sample. However, self-samples cannot be used for cytology assessment, necessitating that women who used self-sampling in this cohort visit a GP if they are found to be HPV-positive. We assumed 79% compliance with the request for cytology in women who use self-sampling [12]. If the cytology result is then abnormal, women are referred to a gynecologist for confirmatory colposcopy (direct referral). If the cytology result is normal, cytology is repeated after 6 months, and women are only referred for confirmatory colposcopy if the repeat cytology is positive (indirect referral). Women with a normal repeat cytology return to the standard cervical cancer screening program, with the next invitation in 5 years.

The following assumptions were made for this base scenario: (1) since the model performs cycles per year, and not every 6 months, repeat cytology is performed at the same time as triage cytology; (2) the sensitivity and specificity of the confirmatory diagnosis was set at 100%; (3) all women with a FIGO diagnosis exit the screening program; and (4) all women with CIN2 or CIN3 are treated, return to hrHPV-negative state, and enter regular screening. The flowchart of the base scenario is presented in the Appendix A. The costs for primary screening, cytology triage, diagnosis, and treatment were taken as reported by Jansen et al., 2021 [8], and the report “The effects and costs of the population screening for cervical cancer in the Netherlands after the restructuring” in 1996 [32]. All costs were indexed to 2021 (Table 3).

#### 2.2.2. Alternative Screening Scenarios

Two different scenarios were simulated to investigate the effects of using molecular triage testing.

In scenario I, a molecular test as triage was applied to all hrHPV-positive women. Those who tested positive for the molecular test were referred for confirmatory colposcopy diagnosis, while those who tested negative returned to screening in 5 or 10 years, depending on their age and hrHPV result (Appendix A).

In scenario II, the molecular test was only applied as triage for hrHPV-positive women who used the self-sampling device, with follow-up the same as in scenario I. Women who underwent hrHPV testing by a GP followed the procedure in the base scenario (Appendix A).

To date, no molecular test has undergone validation of its clinical performance (i.e., sensitivity, specificity, and predictive values) in a large prospective cohort of hrHPV-positive women, with any existing research producing heterogeneous results [13,15,33]. Therefore, we varied the sensitivity and specificity of the molecular test between 65% and 95% (in 10% increments) and performed 10 iterations for each of the 16 possible sensitivity and specificity combinations when calculating the average point estimates. Finally, the average LYs, number of colposcopy referrals, screen-detected CIN2+ states, and false positives were computed over the lifetime for each scenario and presented in a heatmap using the outcomes of the base screening scenario for comparison.

### 2.3. Validation of the Model

Internal and external validation of the model were conducted using the base scenario, which included all input parameters. For internal validation, we plotted the number of women in each state per cycle with and without screening to verify the expected reduction in cervical cancer cases after screening. For the external validation, we simulated 100,000 women and compared the simulated hrHPV prevalence with the observed prevalence in Monitor 2019 [12]. Additionally, we estimated and compared the proportions of CIN2+ and CIN3+ cases detected by direct referral using our SiMCerC model and the data observed in Monitor 2019 [12]. To ensure robust results, we performed 10 iterations to estimate average point estimates.

### 2.4. Headroom Analysis

A headroom analysis was performed to determine how much an intervention can cost to be considered cost-effective, given the associated health benefits and a cost threshold for those benefits [34]. In this study, the maximum reimbursable price (MRP) of molecular triage was assessed. The MRP was defined as the price at which a screening including molecular triage testing is still cost-effective. This MRP represents the potential unit value of a new technique, and was estimated as follows [34]:$$MRP=\left(Net\;reduction\;in\;services\;cost\;excluding\;the\;price\;of\;the\;molecular\;test\right)+\lambda \times YG$$
where λ represents the maximum willingness to pay (MWP) in the Netherlands, set at EUR 20,000 per quality-adjusted life-year (QALY) [35].

For this analysis, we used life-years gained (LYGs) instead of QALYs and maintained the same threshold. The net reduction in service cost was calculated as the difference between the total costs of the alternative scenario and the base scenario per woman. To estimate the total cost of the base scenario, the costs of primary screening, triage (cytology), confirmatory diagnosis, and treatment were included. In the alternative scenario, all costs were set equal to the base scenario, while the cost of molecular triage was not included because this was the one we aimed to estimate. LYGs were calculated as the difference in average LYs between the alternative and base scenarios. The average LYs were calculated by summing the number of women alive in each cycle and then dividing that total by 100,000 (i.e., the cohort size).

### 2.5. Sensitivity Analysis: Effect of Participation Rate Parameters

Three univariate analyses were conducted to explore the effects of participation rate parameters on the MRP of molecular triage testing. For this, the sensitivity and specificity of the molecular triage test were set at 80% and 85%, respectively. First, we compared the MRP when the participation rate changed from 79% to 90% for repeat cytology, because higher participation was expected with the extended follow-up time. Second, to assess the impact of growing self-sampling device use in the Netherlands, we compared the MRP if self-sampling participation grew from 9% to 22% based on national data for the monitoring of cervical cancer screening in the Netherlands (Monitor 2021) [36]. However, note that overall participation per age group remained unchanged, as these rates remained similar between 2019 and 2021. Third, because self-sampling aims to achieve a desired 70% overall primary participation threshold, we compared the MRP when the total participation per age increased to 70% and 30% used self-sampling per round.

## 3. Results

### 3.1. Model Validation

As part of the internal validation, we calculated the total number of women per state per cycle, both with and without screening (Appendix A), to illustrate the expected reduction in cancer state per cycle when screening is implemented. Figure 1 presents the results of a comparison between the simulated data (SiMCerC) and the observed data (Monitor 2019) for HPV prevalence, together with the proportion of CIN2+ and CIN3+ cases detected through direct referrals (external validation). The absolute numbers of CIN2, CIN3, and cancer cases detected in each screening round, per 1000 direct referrals in each age group, were also calculated (Appendix A). The SiMCerC model satisfactorily reproduced both the hrHPV prevalence and the screen-detected CIN2, CIN3, and cancer cases by age group in direct referrals.

### 3.2. Outcomes per Scenario

Figure 2 shows the outcomes generated in each screening scenario in heatmaps: average LYs, total number of colposcopy referrals, screen-detected CIN2+ cases, colposcopy referrals/CIN2+ rate, women referred with a false-positive test, and lifetime costs. The average LYs across the three scenarios was approximately 76. In scenarios I and II, we observed a slight increase in LYs when the sensitivity and specificity were increased, with scenario II showing a more noticeable increase. Scenario I also reduced the number of colposcopy referrals compared with the base scenario due to increased specificity, which decreased the false positives and costs; however, the higher sensitivity increased the detection of CIN2+. Scenario II demonstrated the highest LYGs (0.031) with a sensitivity and specificity of 95%. The numbers of colposcopy referrals, screen-detected CIN2+, colposcopy referrals/CIN2+ rate, and false positives over a lifetime were comparable between the base scenario and scenario II.

### 3.3. Maximum Reimbursable Price

In scenario I, the minimum sensitivity and specificity required for the molecular test to be cost-effective were 85% and 75%, respectively. The corresponding MRP ranged from EUR 244 to EUR 435 (Figure 2). In scenario II, all sensitivity and specificity combinations were deemed cost-effective, with the MRP ranging from EUR 162 to EUR 624 (Figure 2).

### 3.4. Sensitivity Analysis: Effect of Participation Rate Parameters

Figure 3 shows the results of the sensitivity analysis. Using the base scenario for reference, with a sensitivity of 80% and a specificity of 85% for molecular testing, the MRPs for molecular testing were EUR 227 and EUR 314 in scenarios I and II, respectively. Increasing the repeat cytology participation from 79% to 90% in the base scenario led to an MRP reduction in both scenarios (EUR 125 for scenario I and EUR 212 for scenario II). When self-sampling increased from 9% to 22%, the MRP for molecular testing decreased slightly to EUR 202 for scenario I and EUR 288 for scenario II. Finally, when the overall primary participation rate increased to 70%, assuming that 30% of the participants would use self-sampling per round, the MRP decreased to EUR 105 for scenario I and EUR 122 for scenario II.

## 4. Discussion

We developed and validated a microsimulation Markov model for cervical cancer screening (the SiMCerC model), specific to the Netherlands, and used it to estimate the MRP of molecular testing for triaging hrHPV-positive women in an hrHPV-based cervical cancer screening program. The validation process yielded fairly good results compared with observed data for hrHPV prevalence per age and the detection of CIN2+ or CIN3+ in direct referrals. With an MWP set at EUR 20,000, simulated scenario I then revealed that molecular testing must have a minimum sensitivity of 85% and a minimum specificity of 75% to detect CIN2+ and achieve commercial value at an MRP between EUR 244 and EUR 435. By contrast, in scenario II, all possible combinations of sensitivity and specificity (ranging from 65% to 95%) consistently achieved cost-effectiveness, with an MRP ranging from EUR 162 to EUR 624. This latter result was primarily driven by the effectiveness of cytology, which was used as triage in 91% of the population, while avoiding loss to follow-up among women using self-sampling. We observed a tendency for the MRP to increase with increasing specificity, possibly due to the potential cost reductions associated with fewer colposcopy referrals and the lower costs for women referred with a false-positive test, with minimal impact on LYGs.

The SiMCerC model is a micro-simulation model, or an individual level state transition model (STM) [37]. Instead of a cohort-based model, the history and disease progression of individual women can be followed over a lifetime, as well as the option to study the relationship between variation in patient characteristics and model outcomes. In addition, because of the traceability of individual women in the simulation, we were able to simulate the history and disease progression of individual women without screening and in the presence of different scenarios of screening. Because one of the drawbacks of an STM model can be computational and data requirements, we have programmed SiMCerC in C++ (version 6.3, Austin, TX, USA), which is known for its high computational speed and efficiency. A simulation of 100,000 women in SiMCerC took on average 4.09 s to run on a Windows 11 PC (HP Elite Mini 600 G9 Desktop PC, 12th Gen Intel Core i5, clock 2.00 GHz, 64 bits, RAM 16.0 Gb).

For the external validation of our SiMCerC model, we utilized observed data from Monitor 2019 as comparison with real-world clinical data [12]. Model validation without screening was not feasible due to the presence of a long-standing cervical cancer screening program in the Netherlands, which has significantly altered the incidence of cervical cancer since 1976 [38]. Therefore, we used the base screening scenario to replicate hrHPV prevalence with the screen-detected CIN2+ data reported in Monitor 2019 [12,29]. Although Monitor 2020 was published in 2021, these data were significantly affected by several months of program disruption during the COVID-19 pandemic [38,39]. Consequently, data for that year do not accurately represent typical screening outcomes [38]. Monitor 2021 was published later in 2022, with results similar to those in 2019, except for an increase in self-sampling from 9% to 22% between 2019 and 2022 [12,36]. We opted to utilize these data for the univariate sensitivity analyses to account for the rising trend of using self-sampling. Validation of the SiMCerC model revealed fairly good performance compared with Monitor 2019 due to the simulation of hrHPV prevalence and the proportion of screen-detected CIN2+ in direct referrals.

Our findings demonstrated that scenarios I and II had minimal impact on LYGs, even with a sensitivity of 95%. This result was expected given the low mortality of cervical cancer in the Netherlands, which has remained at a maximum of 2.9 per 100,000 women over the past decade [12]. Consequently, the rationale behind implementing a new triage strategy lies in the potential reduction in unnecessary healthcare utilization and associated costs by increasing specificity. We observed a decrease in the cost of services as specificity increased, consistent with decreases in false positives and colposcopy referrals, particularly in scenario I. These results also compare well with the evaluation of the Dutch cervical cancer screening program for 2017–2020, which led to the implementation of cytology plus genotyping in 2022 in an effort to decrease the number of colposcopy referrals [40,41]. Minor differences between the base scenario and scenario II can be attributed to the fact that 91% of the population undergoes screening by GPs, resulting in the effects and costs being influenced more by cytology than by molecular testing. In the sensitivity analysis, increasing the participation rate for self-sampling from 9% to 22% (consistent with Monitor 2021) led to a decrease in the MRP due to a slight increase in the cost of self-sampling, leaving less room for spending on molecular triage.

The MRP was generally higher for scenario II than for scenario I. However, we only estimated the possible cost of a single test, whereas calculating the total development costs requires volume to be considered [42]. Although the MRP was lower for scenario I, this requires a higher volume because all hrHPV-positive women would need molecular testing for triage. By contrast, in scenario II, only hrHPV-positive women within 9% to 22% of self-sampling users would require molecular testing.

The sensitivity analysis revealed a decrease in the MRP as participation in repeat cytology increased, primarily due to increased effects and costs in the base scenario. This outcome was expected because loss to follow-up is a major disadvantage of using cytology as triage [9,10], and this could be resolved if all hrHPV-positive women who performed self-sampling attended triage cytology. Alternatively, increasing overall participation to 70% also decreased the MRP, mainly due to the higher costs associated with self-sampling and less room for spending in molecular triage. In both scenarios, the MRP remained cost-effective with a minimum MRP of EUR 105 when using a sensitivity of 80% and a specificity of 85%.

To our knowledge, no country has implemented molecular testing alone as a triage test for hrHPV-positive women, making it difficult to compare the average costs of such an intervention. As there are no existing studies for direct comparison, this study presents the first headroom analysis for the cost-effectiveness of a molecular triage test in the context of self-sampling and GP-led sampling, given the sensitivity and specificity of molecular triage testing to detect CIN2+ in an hrHPV-based screening program tailored to the Dutch context. However, it is important to acknowledge the limitations of our study.

First, we evaluated health benefits using LYGs, while reporting the cost-effectiveness threshold using QALYs [35]. We chose LYGs as our measure because the primary aim of screening is to reduce mortality, and LYGs were a natural measure of that effect within our lifelong model [43]. Additionally, we did not have utility values available, so we were unable to calculate QALYs. Previous studies have also found no conflict when using LYG or QALY measures [43]. Despite this, QALYs are lower than LYGs, but they are valued more due to representing perfect health. Therefore, MWP for LYG would be lower [43].

Second, there have been recent changes (July 2022) to the cervical cancer screening program in the Netherlands, which has introduced hrHPV genotyping plus cytology for triage [44], and these were not accounted for in our base screening scenario. Nevertheless, incorporating these changes would pose challenges to achieving the outcomes associated with molecular testing alone. As previously mentioned, the use of (single-assay) molecular testing, particularly in conjunction with the increasing use of self-sampling, may contribute to a reduction in the number of visits per woman.

Third, studies assessing molecular testing in hrHPV-positive women present their findings using a CIN3+ threshold [14,15]. However, we could not investigate the effects of solely detecting and treating CIN3+ due to the absence of data on the performance of the program when exclusively treating CIN3. Overall, therefore, while our study sheds light on the cost-effectiveness of molecular triage testing within the Dutch context, further research is needed to explore its applicability in other settings and to account for recent changes in screening protocols.

Finally, our model has some limitations in its assumptions. (a) We assume that it is safe for women who test HPV-positive but negative on triage tests not to receive additional follow-ups. In practice, however, there may be cases where further examination does become necessary to ensure safety. (b) Our model assumes that the probability of screening participation is independent of whether an individual has participated in previous screening rounds. While this assumption simplifies the model, it cannot fully capture the complexity of patient behavior, which may vary over time. (c) We did not consider HPV-type-specific transition probabilities in our model. Although the inclusion of HPV-type-specific data could provide a more nuanced understanding of disease progression, we did not make this addition because we believe it is not necessary to answer our question. (d) We did not consider the age-specific distribution for participation through self-sampling. However, since the overall participation rate for self-sampling was relatively low [12], its influence on our results is negligible. Nevertheless, it is important to consider the age distribution in future analyses if participation in self-sampling increases. Our model focuses on the discriminatory power of CIN2+ under histological progression, and our results provide valuable insights into the MRP of molecular testing.

In the Netherlands, despite the need to reduce the number of women referred with false-positive results, cytology as a triage method has a sensitivity of 78% and a specificity of 80%, indicating good performance [31]. Achieving a molecular test with a specificity of at least 85%, without compromising sensitivity, poses an important challenge [10,14]. However, if this performance could be achieved, potential benefits include reduced follow-up visits for self-sampling users and fewer unnecessary interventions for both the healthcare system and women [40]. It is noteworthy that implementing molecular testing as a triage method may also offer more advantages in countries where the reproducibility and performance of cytology are relatively poor or in settings with limited cytologists. In such settings, relying solely on cytology for triage may introduce variability, increase the loss of follow-up (of women who use self-sampling), and reduce the overall effectiveness of screening [45,46].

## 5. Conclusions

In conclusion, our study highlights that specificity should be a key focus for the development of a molecular triage test in cervical cancer screening. If a specificity of at least 85% and a sensitivity of 80% can be achieved by molecular testing, the maximum costs could range from EUR 227 to EUR 314 per unit. In this scenario, the application of molecular triage testing to all hrHPV-positive women would be more advantageous due to the number of units required for screening. These findings emphasize the potential benefits of implementing molecular testing with higher specificity to facilitate more accurate and efficient screening without losing LYGs. In turn, this could reduce healthcare costs and minimize unnecessary interventions for both the healthcare system and women, ultimately improving the effectiveness of cervical cancer screening programs.

## Figures and Tables

**Figure 1 cancers-17-00612-f001:**
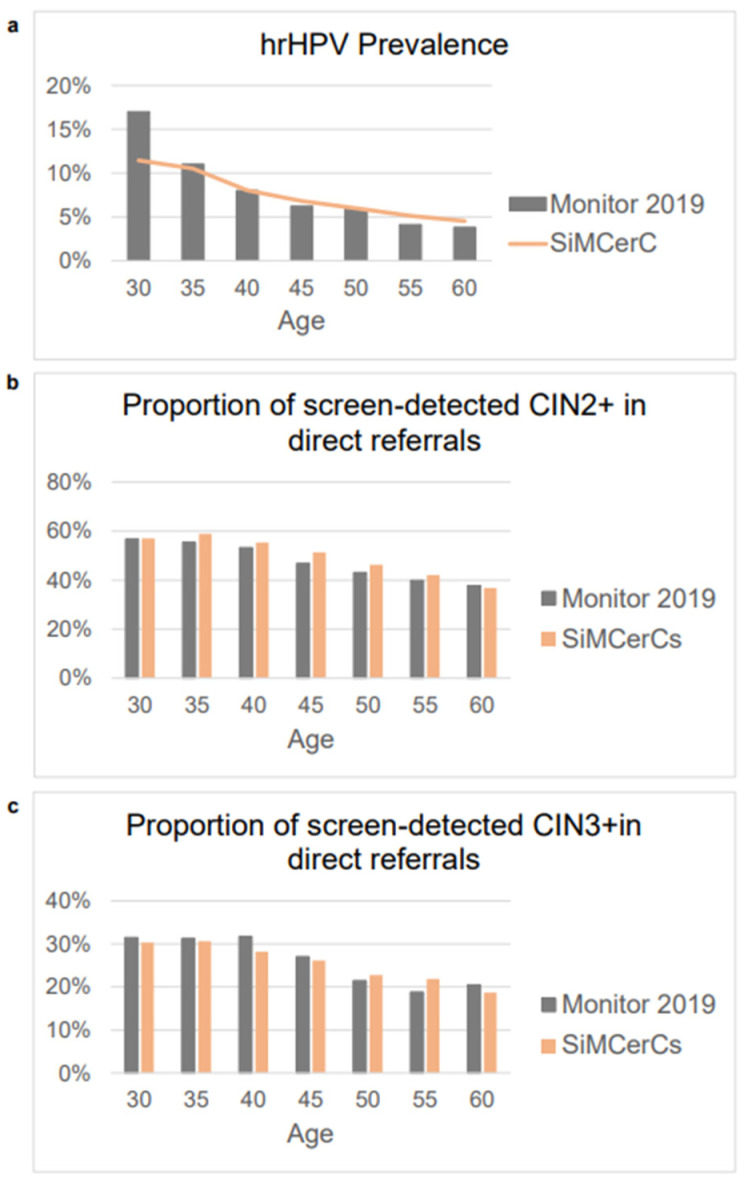
Comparison of hrHPV prevalence (**a**) and the proportion of CIN2+ (**b**) and CIN3+ (**c**) cases detected by direct referral between the SiMCerC simulation and the observations in Monitor 2019. Abbreviations: CIN, cervical intraepithelial neoplasia; hrHPV, high-risk human papillomavirus; SiMCerC, a microsimulation Markov model for cervical cancer screening.

**Figure 2 cancers-17-00612-f002:**
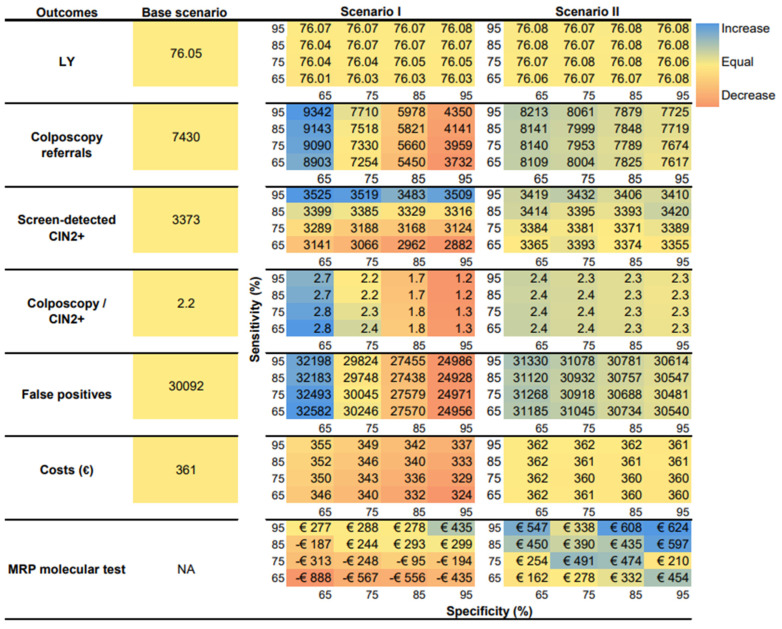
Heatmap of outcomes generated in each screening scenario by sensitivity and specificity. The heatmap uses a scale from orange to blue to represent the smallest and highest values. The darkest orange represents values that decrease the most compared with the base scenario, while the darkest blue represents values that increase the most compared with the base scenario. Yellow represents values that are approximately similar to the reference base scenario. Abbreviations: CIN, cervical intraepithelial neoplasia; LYs, life-years; MRP, maximum reimbursable price.

**Figure 3 cancers-17-00612-f003:**
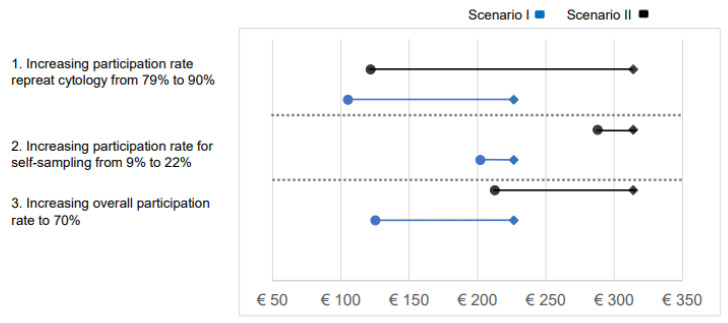
Results of the sensitivity analyses: effect of participation rate parameters. The blue and black diamonds indicate the base MRP of the molecular test for scenarios I and II, respectively. The horizontal lines indicate how the MRP of each scenario would change in three scenarios: (1) if participation for repeat cytology increased from 79% to 90%, (2) if participation through self-sampling increased from 9% to 22%, and (3) if overall participation increased to 70% through self-sampling.

**Table 1 cancers-17-00612-t001:** Transition probabilities per year for the natural history of cervical cancer.

Parameters			Transition Probability (95% CI)	Refs.
From Stage	To Stage	Age (Years)		
hrHPV−	hrHPV+	15–24	0.074 (0.063–0.084)	[18,20,21]
		25–34	0.098 (0.089–0.106)	
		35–44	0.050 (0.045–0.055)	
		45–54	0.036 (0.032–0.040)	
		≥55	0.027 (0.023–0.031)	
hrHPV+	hrHPV−		0.5034 (0.457–0.542)	[22]
	CIN1		0.0610 (0.048–0.075)	[23]
	CIN2		0.0034 (0.002–0.005)	[24]
	CIN3		0.0072 (0.006–0.009)	
CIN1	hrHPV−		0.4000 (0.260–0.540)	[25]
	hrHPV+		0.1600 (0.060–0.260)	
	CIN2		0.0242 (0.010–0.039)	
	CIN3		0.0047 (0–0.011)	
CIN2	hrHPV+		0.0073 (0.003–0.012)	[26]
	CIN1		0.0244 (0.011–0.039)	
	CIN3		0.0474 (0.0289–0.0679)	
CIN3	hrHPV+		0.0025 (0–0.006)	[27]
	CIN1		0.0074 (0.002–0.013)	
	CIN2		0.0486 (0.034–0.064)	
	cancer		0.04 (0.03–0.05)	[28]

Abbreviations: CIN, cervical intraepithelial neoplasia; hrHPV, high-risk human papillomavirus (+, positive; −, negative). Refs.: reference from which the data were extracted.

**Table 2 cancers-17-00612-t002:** Input parameters for cervical cancer screening compliance and clinical performance of the screening test.

Parameter	Values (95% CI)	Refs.
General		[29]
Invitation age	Always at 30, 35, 40, 50, and 60 Invitation at 45, 55, and 65 if the woman was hrHPV+ or did not participate at 40, 50, or 60	
Program attendance and compliance		[12]
Attendance distribution by age		
30 years	44.1%	
35 years	49.3%	
40 years	56.0%	
45 years	58.7%	
50 years	60.4%	
55 years	60.9%	
60 years	61.3%	
GP-collected samples	91%	
Self-sampling collected samples	9%	
Compliance		
with cytology triage (after hrHPV+ by self-sampling)	79.0%	
with direct confirmatory colposcopy	70.0%	
with repeat cytology	63.0%	
with indirect confirmatory colposcopy	58.0%	
Clinical Performance (hrHPV)		[30]
hrHPV test by GP		
Sensitivity	96.4 (92.9–99.9)	
Specificity	94.2 (93.6–94.8)	
hrHPV test by SS		
Sensitivity	92.9 (87.3–98.4)	
Specificity	93.9 (93.4–94.5)	
Clinical Performance (Cytology)		[31]
Cytology as triage		
Sensitivity for CIN2+	78.2 (77.2–79.1)	
Specificity for CIN2+	80.4 (80.0–80.8)	
Control cytology after 6 months		
Sensitivity for CIN2+	86.4 (84.7–88.1)	
Specificity for CIN2+	87.4 (87.0–87.8)	

Abbreviations: CIN, cervical intraepithelial neoplasia; GP, general practitioner; hrHPV, high-risk human papillomavirus (+, positive). Refs.: reference from which the data were extracted.

**Table 3 cancers-17-00612-t003:** Costs of primary screening, cytology triage, diagnosis, and treatment.

Test	Costs	Refs.
Primary screening/triage		[8]
Primary hrHPV by GP	EUR 0.34	
Primary hrHPV by self-sampling	EUR 105.07	
Triage cytology by GP	EUR 27.05	
Triage cytology by self-sampling	EUR 54.10	
Repeat cytology	EUR 55.14	
Confirmatory diagnosis/treatment		[32]
Negative	EUR 344.85	
CIN1	EUR 1073.60	
CIN2	EUR 1588.92	
CIN3	EUR 1860.90	
FIGO 1A	EUR 6094.13	
FIGO 1B	EUR 14,451.27	
FIGO 2+	EUR 14,244.36	

Costs were indexed to 2021. Abbreviations: CIN, cervical intraepithelial neoplasia; FIGO, International Federation of Gynecology and Obstetrics; GP, general practitioner; hrHPV, high-risk human papillomavirus. Refs.: reference from which the data were extracted.

## Data Availability

The data supporting the findings of this study are available within the article, Appendix A, and SiMCerC, accessible at https://github.com/MelisaCastaneda/SiMCerC (Created on 1 May 2024).

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
