# Peer review of "Molecular Testing as Triage in Cervical Cancer Screening: Economic Evaluation Using Headroom Analysis"

_cancers, 2025, doi:10.3390/cancers17040612_

Round 1
Reviewer 1 Report
Comments and Suggestions for Authors
Manuscript “Molecular testing as triage in cervical cancer screening: eco- 2 nomic evaluation using headroom analysis” by Castaneda et al. is an interesting research work. The author conducted detailed experiments and presented data systematically. Results also discussed adequately. However, the following major concerns are associated with the current version of the manuscript. Hence, the author need to revised the manuscript through addressing the following minor comments to improve the overall standard of the manuscript.
Comments to authors
Abbreviations are not disclosed in the text, for example CCIN2 and CIN2+ so on
“what are the recent developments in this direction of research and how this study is overcoming the limitations of previous research” that need to be discussed in introduction
The current introduction is very concise. Slightly elaborate is mandatory
Figure 1. it is suggested to incorporate color figure instead of black & white.
Conclusions need to be elaborated slightly and include how this study will impact on future research directions
There are three tables. It seems there is a scope to present these tables in the form of different types of figures
It is wonder to see the references even in Table 2 and Table 3.
Request to author cite the most recent references in the field of study
Comparative analysis of this study results with other research work published in literature
In Figure 2, the author used different colors to highlight the data. Is there any correction between color and respective data. If there is then it is recommended to mention note at bottom of the Figure 2.
In Figu 1 there are 3 sub-figures and detailed are mentioned on top of the reach figure. It si suggested to label sub-figures as a, b, and c. the in caption mentioned the details of fitures
Reviewer 2 Report
Comments and Suggestions for Authors
Nice study, with some clumsiness when it comes to methodology.
LYG as an indicator of effectiveness is a bit of an unorthodox choice. Authors try to defend it using Robberstad paper from 2005. Maybe they could try to estimate the values of utility and then estimate the impact of various utility values on results using sensitivity analysis.
Another aspect is a more thorough discussion of cohort-level versus patient-level Markov models. Authors could use Gibson's paper to that end as well as recent publications (authors could simply search Scopus within "Article title" using string "patient-level AND markov"). In this way, they will be able discuss the assumptions of their model in more detail.
Authors use WTP in the text without introducing the acronym (willingness-to-pay); they do use MWP (maximum willingness-to-pay). Also in Line 276 they state: "With a WTP set at €20,000, simulated...". WTP can be set at €20,000/QALY, or as in the study at €20,000/LYG. As mentioned it would be beneficial if authors add utility values to the model.
Comments on the Quality of English Language
English could be a bit improved as always, so some effort would be appreciated.
Round 2
Reviewer 1 Report
Comments and Suggestions for Authors
Revised manuscript suitable for publication as author addressed all the comments of reviewer thoroughly